# A biologically plausible neural network for Slow Feature Analysis

**David Lipshutz**[*1]        **Charlie Windolf**[*1,2]        **Siavash Golkar**[1]        **Dmitri B. Chklovskii**[1,3]

[1] Center for Computational Neuroscience, Flatiron Institute
[2] Department of Statistics, Columbia University
[3] Neuroscience Institute, NYU Medical Center

```
{dlipshutz,sgolkar,dchklovskii}@flatironinstitute.org
                 c.windolf@columbia.edu
```

## Abstract

Learning latent features from time series data is an important problem in both machine learning and brain function. One approach, called Slow Feature Analysis (SFA), leverages the slowness of many salient features relative to the rapidly varying input signals. Furthermore, when trained on naturalistic stimuli, SFA reproduces interesting properties of cells in the primary visual cortex and hippocampus, suggesting that the brain uses temporal slowness as a computational principle for learning latent features. However, despite the potential relevance of SFA for modeling brain function, there is currently no SFA algorithm with a biologically plausible neural network implementation, by which we mean an algorithm operates in the online setting and can be mapped onto a neural network with local synaptic updates. In this work, starting from an SFA objective, we derive an SFA algorithm, called Bio-SFA, with a biologically plausible neural network implementation. We validate Bio-SFA on naturalistic stimuli.

## 1   Introduction

Unsupervised learning of meaningful latent features from noisy, high-dimensional data is a fundamental problem for both machine learning and brain function. Often, the relevant features in an environment (e.g., objects) vary on relatively slow timescales when compared to noisy sensory data (e.g., the light intensity measured by a single receptor in the retina). Therefore, temporal slowness has been proposed as a computational principle for extracting relevant latent features [8, 19, 31].

A popular approach for extracting slow features, introduced by Wiskott and Sejnowski [31], is Slow Feature Analysis (SFA). SFA is an unsupervised learning algorithm that extracts the slowest projection, in terms of discrete time derivative, from a nonlinear expansion of the input signal. When trained on natural image sequences, SFA extracts features that resemble response properties of complex cells in early visual processing [2]. Impressively, hierarchical networks of SFA trained on simulated rat visual streams learn representations of position and orientation similar to representations encoded in the hippocampus [9].

The relevance of SFA is strengthened by its close relationship to information theoretic objectives and its equivalence to other successful algorithms under certain assumptions. When the time series is reversible and Gaussian, (Linear) SFA is equivalent to maximizing mutual information between the current output of the system and the next input [7, 5]. Moreover, features extracted by several

---

[*]Equal contribution

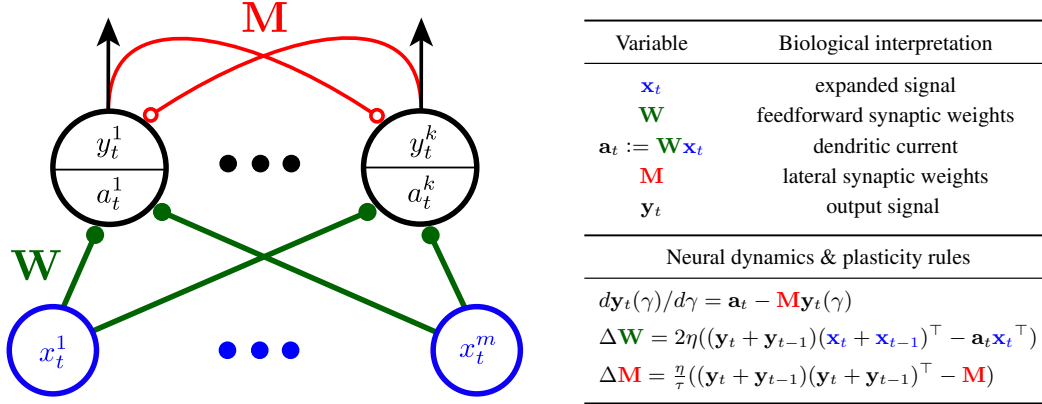

| Variable | Biological interpretation |
|---|---|
| $\mathbf{x}_t$ | expanded signal |
| $\mathbf{W}$ | feedforward synaptic weights |
| $\mathbf{a}_t := \mathbf{W}\mathbf{x}_t$ | dendritic current |
| $\mathbf{M}$ | lateral synaptic weights |
| $\mathbf{y}_t$ | output signal |

| Neural dynamics & plasticity rules |
|---|
| $d\mathbf{y}_t(\gamma)/d\gamma = \mathbf{a}_t - \mathbf{M}\mathbf{y}_t(\gamma)$ |
| $\Delta \mathbf{W} = 2\eta((\mathbf{y}_t + \mathbf{y}_{t-1})(\mathbf{x}_t + \mathbf{x}_{t-1})^\top - \mathbf{a}_t\mathbf{x}_t^\top)$ |
| $\Delta \mathbf{M} = \frac{\eta}{\tau}((\mathbf{y}_t + \mathbf{y}_{t-1})(\mathbf{y}_t + \mathbf{y}_{t-1})^\top - \mathbf{M})$ |

Figure 1: A biologically plausible neural network implementation of Bio-SFA. The figure on the left depicts the architecture of the neural network. Blue circles are the input neurons and black circles are the output neurons with separate dendritic and somatic compartments. Lines with circles connecting the neurons denote synapses. Filled (resp. empty) circles denote non-Hebbian (resp. anti-Hebbian) synapses.

algorithms favoring predictability from real-world datasets are similar to those extracted by SFA [29]. Finally, (Linear) SFA is equivalent to a time-lagged independent components analysis [3, 10], which is a popular statistical technique used to analyze molecular dynamics [22, 20, 26, 27].

Due to its success in modeling aspects of neural processing, deriving an algorithm for SFA with a biologically plausible neural network implementation is an important task. For the purposes of this work, we define biologically plausible to mean that the neural network operates in the online setting (i.e., after receiving an input, it computes its output before receiving its next input, never storing a significant fraction of past inputs), and its synaptic learning rules are local (i.e., a synaptic weight update depends only on variables represented in the pre- and postsynaptic neurons). In addition to satisfying basic properties of neural circuits, these online and locality requirements can lead to networks that are well-suited for analyzing large datasets because they operate in the online setting with low computational overhead.

While there are a few online algorithms for SFA, none have biologically plausible neural network implementations that extract multiple slow features. Moreover, there are no neural network implementations for the related information theoretic algorithms discussed above [29, 5]. Kompella et al. propose Incremental SFA [14] (see [16, 32] for extensions). However, this approach relies on non-local learning rules, so it does not meet the above criteria for biological plausibility. Malik et al. [17] use an online generalized eigenvalue problem solver [33] to derive an online algorithm for SFA. While their algorithm for finding one-dimensional projections can be implemented in a biologically plausible network, their extension to multi-dimensional projections is not fully online.

In this work, we propose Bio-SFA: an online algorithm for SFA with a biologically plausible neural network implementation, Fig. 1. We adopt a normative approach to derive our algorithm. First, we express the solution of the SFA problem in terms of an objective from classical multidimensional scaling. We then manipulate the objective to arrive at a min-max optimization problem that can be solved in the online setting by taking stochastic gradient descent-ascent steps. These steps can be expressed in terms of neural activities and updates to synaptic weight matrices, which leads to a natural interpretation of our online algorithm as a biologically plausible neural network. To validate our approach, we test our algorithm on datasets of naturalistic stimuli and reproduce results originally performed in the offline setting.

The synaptic updates of the feedforward weights $\mathbf{W}$ in our network are similar, although not identical, to the updates proposed heuristically by Földiák [8] to extract slow temporal features. However, there is no theoretical analysis of the algorithm in [8]. In contrast, in our normative approach, Bio-SFA is derived directly from an SFA objective, so we can analytically predict its output, as well as the synaptic weights, without resorting to numerical simulation. In addition, the comparison of our learning rules with Földiák's illuminates the relationship of [8] to SFA.

## 2 Slow Feature Analysis

Here and below, vectors are boldface lowercase letters (e.g., $\mathbf{v}$), and matrices are boldface uppercase letters (e.g., $\mathbf{M}$). We use superscripts to denote the components of a vector (e.g., $v^i$).

### 2.1 Problem statement

Wiskott and Sejnowski [31] proposed the following 2 step method for extracting slow features from a noisy data set: (1) generate a nonlinear expansion of the input signal, and (2) find the slowest, in terms of discrete time derivative, low-dimensional projection of the expanded signal. In this section, we review these 2 steps.

Let $\{\mathbf{s}_0, \mathbf{s}_1, \ldots, \mathbf{s}_T\}$ be a $d$-dimensional input signal.[2] The first step of SFA is to generate an $m$-dimensional expansion $\{\mathbf{x}_t\}$, referred to as the expanded signal, of $\{\mathbf{s}_t\}$. Let $\mathbf{h} = (h^1, \ldots, h^m) : \mathbb{R}^d \to \mathbb{R}^m$ be an expansion function and define

$$\mathbf{x}_t := \mathbf{h}(\mathbf{s}_t) - \frac{1}{T} \sum_{t'=1}^{T} \mathbf{h}(\mathbf{s}_{t'}), \qquad t = 0, 1, \ldots, T,$$

so that $\{\mathbf{x}_t\}$ is centered.

Let $k < m$. The second step of SFA is to find the $k$-dimensional linear projection $\{\mathbf{y}_t\}$ of the expanded signal $\{\mathbf{x}_t\}$ that minimizes the mean discrete-time derivative of the output signal $\{\mathbf{y}_t\}$, subject to a whitening constraint. To be precise, the objective can be formulated as follows:

$$\underset{\{\mathbf{y}_t\}}{\arg\min} \frac{1}{T} \sum_{t=1}^{T} \|\dot{\mathbf{y}}_t\|^2 \quad \text{subject to} \quad \frac{1}{T} \sum_{t=1}^{T} \mathbf{y}_t \mathbf{y}_t^\top = \mathbf{I}_k, \tag{1}$$

where $\dot{\mathbf{y}}_t$ is the discrete time derive of $\mathbf{y}_t$, and $\mathbf{y}_t$ is a linear projection of $\mathbf{x}_t$; that is,

$$\dot{\mathbf{y}}_t := \mathbf{y}_t - \mathbf{y}_{t-1}, \qquad t = 1, \ldots, T, \tag{2}$$

$$\mathbf{y}_t := \mathbf{V}^\top \mathbf{x}_t, \qquad t = 0, 1, \ldots, T, \qquad \text{for some } \mathbf{V} \in \mathbb{R}^{m \times k}. \tag{3}$$

Note, since $\{\mathbf{x}_t\}$ is centered, the projection $\{\mathbf{y}_t\}$ is also centered.

### 2.2 Quadratic SFA

The focus of this work is to derive a biologically plausible neural network that learns to output the optimal output signal $\{\mathbf{y}_t\}$ when streamed the expanded signal $\{\mathbf{x}_t\}$. While our algorithm does not depend on the specific choice of the expansion function $\mathbf{h}$, for concreteness, we provide an example here.

In their original paper, Wiskott and Sejnowski [31] proposed setting the components of the function $\mathbf{h} : \mathbb{R}^d \to \mathbb{R}^m$ to be the monomials of degree one and two. This choice, which we refer to as "Quadratic SFA", has been widely used in applications [31, 2, 9, 34]. In particular, let $m := d + d(d+1)/2$ and $h^1, \ldots, h^m : \mathbb{R}^d \to \mathbb{R}$ denote the $m$ possible linear and quadratic functions of the form

$$h(\mathbf{s}) := s^i \qquad \text{or} \qquad h(\mathbf{s}) := s^i s^j,$$

for $1 \le i \le j \le d$. (When only the linear features are used, i.e., $x^i = s^i + \text{const}$, this is referred to "Linear SFA".) Thus, each component of the output signal is a quadratic polynomial in the components of the signal of the form:

$$y^i = V_{1i} h^1(\mathbf{s}) + \cdots + V_{mi} h^m(\mathbf{s}) + \text{const}. \tag{4}$$

Biologically, there are a number of mechanism that have been proposed for computing products of the form $s^i s^j$; see, e.g., [13] and the references therein. One such mechanism uses "Sigma-Pi" units [23], which multiplies two inputs via gating and have been invoked in cortical modeling [18].

In Sec. 6, we perform our numerical experiments using the quadratic expansion.

## 3  A novel SFA objective from classical multidimensional scaling

To derive an SFA network, we identify an objective function whose optimization leads to an online algorithm that can be implemented in a biologically plausible network. To identify the objective function, we first rewrite the SFA output as a principal subspace projection and then take advantage of the fact that principal subspace projections can be expressed as solutions of objectives from classical multidimensional scaling [6]. This approach is similar to the derivation of a biologically plausible neural network for canonical correlation analysis [15].

To begin, we define the discrete derivative process $\{\dot{\mathbf{x}}_t\}$ and the delayed sum process $\{\bar{\mathbf{x}}_t\}$ by $\dot{\mathbf{x}}_t := \mathbf{x}_t - \mathbf{x}_{t-1}$ and $\bar{\mathbf{x}}_t := \mathbf{x}_t + \mathbf{x}_{t-1}$, for $t = 1, \ldots, T$. In addition, we define the sample covariance matrices

$$\mathbf{C}_{xx} := \frac{1}{T}\sum_{t=1}^{T}\mathbf{x}_t\mathbf{x}_t^\top, \qquad \mathbf{C}_{\dot{x}\dot{x}} := \frac{1}{T}\sum_{t=1}^{T}\dot{\mathbf{x}}_t\dot{\mathbf{x}}_t^\top, \qquad \mathbf{C}_{\bar{x}\bar{x}} := \frac{1}{T}\sum_{t=1}^{T}\bar{\mathbf{x}}_t\bar{\mathbf{x}}_t^\top. \tag{5}$$

Substituting the definitions in Eqs. (2), (3) and (5) into the objective in Eq. (1), we can equivalently write the SFA problem as the following constrained minimization problem of the projection matrix $\mathbf{V}$:

$$\underset{\mathbf{V}\in\mathbb{R}^{m\times k}}{\arg\min}\ \operatorname{Tr}\mathbf{V}^\top\mathbf{C}_{\dot{x}\dot{x}}\mathbf{V} \quad \text{subject to} \quad \mathbf{V}^\top\mathbf{C}_{xx}\mathbf{V} = \mathbf{I}_k. \tag{6}$$

Due to the whitening constraint in Eq. (6), we can equivalently write it as the maximization of the one-step autocorrelation of the projection $\{\mathbf{y}_t\}$ (see Appendix A for details):

$$\underset{\mathbf{V}\in\mathbb{R}^{m\times k}}{\arg\max}\ \operatorname{Tr}\mathbf{V}^\top\mathbf{C}_{\bar{x}\bar{x}}\mathbf{V} \quad \text{subject to} \quad \mathbf{V}^\top\mathbf{C}_{xx}\mathbf{V} = \mathbf{I}_k. \tag{7}$$

Next, setting $\hat{\mathbf{x}}_t := \mathbf{C}_{xx}^{-1/2}\bar{\mathbf{x}}_t$ for $t = 1, \ldots, T$, and

$$\hat{\mathbf{V}} := \mathbf{C}_{xx}^{1/2}\mathbf{V}, \qquad \mathbf{C}_{\hat{x}\hat{x}} := \frac{1}{T}\sum_{t=1}^{T}\hat{\mathbf{x}}_t\hat{\mathbf{x}}_t^\top = \mathbf{C}_{xx}^{-1/2}\mathbf{C}_{\bar{x}\bar{x}}\mathbf{C}_{xx}^{-1/2},$$

we see that $\mathbf{V}$ is a solution of Eq. (7) if and only if $\hat{\mathbf{V}}$ is the solution of:

$$\underset{\hat{\mathbf{V}}\in\mathbb{R}^{m\times k}}{\arg\max}\ \operatorname{Tr}\hat{\mathbf{V}}^\top\mathbf{C}_{\hat{x}\hat{x}}\hat{\mathbf{V}} \quad \text{subject to} \quad \hat{\mathbf{V}}^\top\hat{\mathbf{V}} = \mathbf{I}_k. \tag{8}$$

Notably, Eq. (8) is the variance maximization objective for the PCA eigenproblem, which is optimized when the column vectors of $\hat{\mathbf{V}}$ span the $k$-dimensional principal subspace of $\mathbf{C}_{\hat{x}\hat{x}}$.

Finally, we take advantage of the fact that principal subspace projections can be expressed as solutions of objectives from classical multidimensional scaling [6, 21]. To this end, define the data matrices

$$\bar{\mathbf{X}} := [\bar{\mathbf{x}}_t, \ldots, \bar{\mathbf{x}}_T], \qquad \hat{\mathbf{X}} := [\hat{\mathbf{x}}_1, \ldots, \hat{\mathbf{x}}_T], \qquad \bar{\mathbf{Y}} := [\bar{\mathbf{y}}_1, \ldots, \bar{\mathbf{y}}_T].$$

Then, since $\bar{\mathbf{y}}_t = \mathbf{V}^\top\bar{\mathbf{x}}_t = \hat{\mathbf{V}}^\top\hat{\mathbf{x}}_t$, we see that $\bar{\mathbf{Y}}$ is the projection of $\hat{\mathbf{X}}_t$ onto its $k$-dimensional principal subspace. As shown in [6], this principal projection can be expressed as a solution of the following objective from classical multidimensional scaling:

$$\underset{\bar{\mathbf{Y}}\in\mathbb{R}^{k\times T}}{\arg\min}\ \frac{1}{2T^2}\left\|\bar{\mathbf{Y}}^\top\bar{\mathbf{Y}} - \hat{\mathbf{X}}^\top\hat{\mathbf{X}}\right\|_{\text{Frob}}^2 = \underset{\bar{\mathbf{Y}}\in\mathbb{R}^{k\times T}}{\arg\min}\ \frac{1}{2T^2}\left\|\bar{\mathbf{Y}}^\top\bar{\mathbf{Y}} - \bar{\mathbf{X}}^\top\mathbf{C}_{xx}^{-1}\bar{\mathbf{X}}\right\|_{\text{Frob}}^2. \tag{9}$$

This objective minimizes the difference between the similarity of consecutive sums of output pairs, $\bar{\mathbf{y}}_t^\top\bar{\mathbf{y}}_{t'}$, and the similarity of consecutive sums of whitened input pairs, $\hat{\mathbf{x}}_t^\top\hat{\mathbf{x}}_{t'}$, where similarity is measured in terms of inner products. Here we have assumed that $\mathbf{C}_{xx}$ is full rank. If $\mathbf{C}_{xx}$ is not full rank (but is at least rank $k$), we can replace $\mathbf{C}_{xx}^{-1}$ in Eq. (9) with the Moore-Penrose inverse $\mathbf{C}_{xx}^+$ (see Appendix A).

## 4  Derivation of an online algorithm

While the objective (9) can be minimized by taking gradient descent steps in $\bar{\mathbf{Y}}$, this does not lead to an online algorithm because the gradient steps require combining inputs from different time steps. Instead, we rewrite the objective as a min-max problem that can be solved by taking gradient descent-ascent steps that correspond to neural activities and synaptic update rules.

## 4.1 A min-max formulation

Expanding the square in Eq. (9) and dropping terms that do not depend on $\bar{\mathbf{Y}}$, we obtain the minimization problem

$$\min_{\bar{\mathbf{Y}} \in \mathbb{R}^{k \times T}} \frac{1}{2T^2} \operatorname{Tr} \left( \bar{\mathbf{Y}}^\top \bar{\mathbf{Y}} \bar{\mathbf{Y}}^\top \bar{\mathbf{Y}} - 2 \bar{\mathbf{Y}}^\top \bar{\mathbf{Y}} \bar{\mathbf{X}}^\top \mathbf{C}_{xx}^{-1} \bar{\mathbf{X}} \right). \tag{10}$$

By introducing dynamical matrix variables $\mathbf{W}$ and $\mathbf{M}$, which will correspond to synaptic weights, we can rewrite the minimization problem (10) as a min-max problem:

$$\min_{\bar{\mathbf{Y}} \in \mathbb{R}^{k \times T}} \min_{\mathbf{W} \in \mathbb{R}^{k \times n}} \max_{\mathbf{M} \in \mathcal{S}_{++}^k} L(\mathbf{W}, \mathbf{M}, \bar{\mathbf{Y}}),$$

where $\mathcal{S}_{++}^k$ denotes the set of $k \times k$ positive definite matrices and

$$L(\mathbf{W}, \mathbf{M}, \bar{\mathbf{Y}}) := \frac{1}{T} \operatorname{Tr} \left( \bar{\mathbf{Y}}^\top \mathbf{M} \bar{\mathbf{Y}} - 2 \bar{\mathbf{Y}}^\top \mathbf{W} \bar{\mathbf{X}} \right) - \operatorname{Tr} \left( \frac{1}{2} \mathbf{M}^2 - \mathbf{W} \mathbf{C}_{xx} \mathbf{W}^\top \right). \tag{11}$$

This step can be verified by differentiating $L(\mathbf{W}, \mathbf{M}, \bar{\mathbf{Y}})$ with respect to $\mathbf{W}$ and $\mathbf{M}$ and noting that the optimal values are achieved when $\mathbf{W}$ and $\mathbf{M}$ equal $\frac{1}{T} \bar{\mathbf{Y}} \bar{\mathbf{X}}^\top \mathbf{C}_{xx}^{-1}$ and $\frac{1}{T} \bar{\mathbf{Y}} \bar{\mathbf{Y}}^\top$, respectively. Finally, we interchange the order of minimization with respect to $\bar{\mathbf{Y}}$ and $\mathbf{W}$, as well as the order of optimization with respect to $\bar{\mathbf{Y}}$ and with respect to $\mathbf{M}$:

$$\min_{\mathbf{W} \in \mathbb{R}^{k \times m}} \max_{\mathbf{M} \in \mathcal{S}_{++}^k} \min_{\bar{\mathbf{Y}} \in \mathbb{R}^{k \times T}} L(\mathbf{W}, \mathbf{M}, \bar{\mathbf{Y}}). \tag{12}$$

The second interchange is justified by the fact that $L(\mathbf{W}, \mathbf{M}, \bar{\mathbf{Y}})$ satisfies the saddle point property with respect to $\bar{\mathbf{Y}}$ and $\mathbf{M}$, which follows from the fact that $L(\mathbf{W}, \mathbf{M}, \bar{\mathbf{Y}})$ is strictly convex in $\bar{\mathbf{Y}}$ (since $\mathbf{M}$ is positive definite) and strictly concave in $\mathbf{M}$.

## 4.2 Offline algorithm

In the offline, or batch, setting, we have access to the sample covariance matrices $\mathbf{C}_{xx}$ and $\mathbf{C}_{\bar{x}\bar{x}}$, and we solve the min-max problem (12) by alternating optimization steps. First, for fixed $\mathbf{W}$ and $\mathbf{M}$, we minimize the objective function $L(\mathbf{W}, \mathbf{M}, \bar{\mathbf{Y}})$ over $\bar{\mathbf{Y}}$, to obtain

$$\bar{\mathbf{Y}} = \mathbf{M}^{-1} \mathbf{W} \bar{\mathbf{X}}. \tag{13}$$

With $\bar{\mathbf{Y}}$ fixed, we then perform a gradient descent-ascent step with respect to $\mathbf{W}$ and $\mathbf{M}$:

$$\mathbf{W} \leftarrow \mathbf{W} + 2\eta \left( \frac{1}{T} \bar{\mathbf{Y}} \bar{\mathbf{X}}^\top - \mathbf{W} \mathbf{C}_{xx} \right) \tag{14}$$

$$\mathbf{M} \leftarrow \mathbf{M} + \frac{\eta}{\tau} \left( \frac{1}{T} \bar{\mathbf{Y}} \bar{\mathbf{Y}}^\top - \mathbf{M} \right). \tag{15}$$

Here $\tau > 0$ is the ratio of the learning rates of $\mathbf{W}$ and $\mathbf{M}$ and $\eta \in (0, \tau)$ is the (possibly time-dependent) learning rate for $\mathbf{W}$. The condition $\eta < \tau$ ensures that matrix $\mathbf{M}$ remains positive definite given a positive definite initialization.

## 4.3 Online algorithm

In the online setting, the expanded signal $\{\mathbf{x}_t\}$ is streamed one sample at a time, and the algorithm must compute its output without storing any significant fraction of the data in memory. In this case, at each time-step $t$, we compute the output $\mathbf{y}_t = \mathbf{M}^{-1} \mathbf{a}_t$, where $\mathbf{a}_t := \mathbf{W} \mathbf{x}_t$ is the projection of $\mathbf{x}_t$ onto the $k$-dimensional "slow" subspace, in a biologically plausible manner by running the following fast (neural) dynamics to equilibrium (our algorithm implements these dynamics using an Euler approximation):

$$\frac{d\mathbf{y}_t(\gamma)}{d\gamma} = \mathbf{a}_t - \mathbf{M} \mathbf{y}_t(\gamma). \tag{16}$$

To update the (synaptic) matrices $\mathbf{W}$ and $\mathbf{M}$, we replace the covariance matrices in (14)–(15) with the rank-1 stochastic approximations:

$$\frac{1}{T}\bar{\mathbf{Y}}\bar{\mathbf{X}}^\top \mapsto \bar{\mathbf{y}}_t\bar{\mathbf{x}}_t^\top, \qquad\qquad \frac{1}{T}\bar{\mathbf{Y}}\bar{\mathbf{Y}}^\top \mapsto \bar{\mathbf{y}}_t\bar{\mathbf{y}}_t^\top, \qquad\qquad \mathbf{C}_{xx} \mapsto \mathbf{x}_t\mathbf{x}_t^\top.$$

This yields the following stochastic gradient descent-ascent steps with respect to $\mathbf{W}$ and $\mathbf{M}$:

$$\mathbf{W} \leftarrow \mathbf{W} + 2\eta\left(\bar{\mathbf{y}}_t\bar{\mathbf{x}}_t^\top - \mathbf{a}_t\mathbf{x}_t^\top\right)$$

$$\mathbf{M} \leftarrow \mathbf{M} + \frac{\eta}{\tau}\left(\bar{\mathbf{y}}_t\bar{\mathbf{y}}_t^\top - \mathbf{M}\right).$$

We can now state our online SFA algorithm, which we refer to as Bio-SFA (Alg. 1).

---

**Algorithm 1:** Bio-SFA

---

**input** expanded signal $\{\mathbf{x}_0, \mathbf{x}_1, \ldots, \mathbf{x}_T\}$; dimension $k$; parameters $\gamma, \eta, \tau$
**initialize** matrix $\mathbf{W}$ and positive definite matrix $\mathbf{M}$
**for** $t = 1, 2, \ldots, T$ **do**
   $\mathbf{a}_t \leftarrow \mathbf{W}\mathbf{x}_t$                                        ▷ project inputs
   **repeat**
      $\mathbf{y}_t \leftarrow \mathbf{y}_t + \gamma(\mathbf{a}_t - \mathbf{M}\mathbf{y}_t)$                       ▷ compute neural output
   **until** convergence
   $\bar{\mathbf{x}}_t \leftarrow \mathbf{x}_t + \mathbf{x}_{t-1}$
   $\bar{\mathbf{y}}_t \leftarrow \mathbf{y}_t + \mathbf{y}_{t-1}$
   $\mathbf{W} \leftarrow \mathbf{W} + 2\eta(\bar{\mathbf{y}}_t\bar{\mathbf{x}}_t^\top - \mathbf{a}_t\mathbf{x}_t^\top)$                  ▷ synaptic updates
   $\mathbf{M} \leftarrow \mathbf{M} + \frac{\eta}{\tau}(\bar{\mathbf{y}}_t\bar{\mathbf{y}}_t^\top - \mathbf{M})$
**end for**

---

## 5 Biologically plausible neural network implementation

We now demonstrate that Bio-SFA can be implemented in a biologically plausible network, depicted in Fig. 1. Recall that we define a network to be biologically plausible if it computes its output in the online setting and has local learning rules. The neural network consists of an input layer of $m$ neurons (blue circles) and an output layer of $k$ neurons with separate dendritic and somatic compartments (black circles with 2 compartments). At each time $t$, the $m$-dimensional expanded signal $\mathbf{x}_t$, which is represented by the activity of the input neurons, is multiplied by the weight matrix $\mathbf{W}$, which is encoded by the feedforward synapses connecting the input neurons to the output neurons (green lines). This yields the $k$-dimensional projection $\mathbf{a}_t = \mathbf{W}\mathbf{x}_t$, which is represented in the dendritic compartment of the output neurons and then propagated to the somatic compartments. This is followed by the fast recurrent neural dynamics Eq. (16) amongst the somatic compartments of the output neurons, where the matrix $\mathbf{M}$ is encoded by the lateral synapses connecting the layer of output neurons (red lines). These fast neural dynamics equilibrate at $\mathbf{y}_t = \mathbf{M}^{-1}\mathbf{a}_t$. The $k$-dimensional output signal $\mathbf{y}_t$ is represented by the activity of the output neurons.

The synaptic updates are as follows. Recall that $\bar{\mathbf{x}}_t = \mathbf{x}_t + \mathbf{x}_{t-1}$ (resp. $\bar{\mathbf{y}}_t = \mathbf{y}_t + \mathbf{y}_{t-1}$) is the delayed sum of the inputs (resp. outputs), which we assume are represented in the $m$ input neurons (resp. $k$ output neurons). Biologically, they can be represented by slowly changing concentrations (e.g., calcium) at the pre- and post-synaptic terminals. We can write the elementwise synaptic updates in Alg. 1 as

$$W_{ij} \leftarrow W_{ij} + 2\eta\left(\bar{y}_t^i\bar{x}_t^j - a_t^i x_t^j\right), \qquad\qquad 1 \le i \le k,\ 1 \le j \le d, \qquad (17)$$

$$M_{ij} \leftarrow M_{ij} + \frac{\eta}{\tau}\left(\bar{y}_t^i\bar{y}_t^j - M_{ij}\right), \qquad\qquad 1 \le i, j \le k. \qquad (18)$$

Since the $j^{\text{th}}$ input neuron stores the variables $x_t^j, \bar{x}_t^j$ and the $i^{\text{th}}$ output neuron stores the variables $a_t^i, y_t^i, \bar{y}_t^i$, the update for each synapse is local.

It is worth comparing the derived updates to the feedforward weights Eq. (17) to the updates proposed by Földiák [8], which are given by

$$w_{ij} \leftarrow w_{ij} + \eta\left(\bar{y}_t^i x_t^j - \bar{y}_t^i w_{ij}\right), \qquad\qquad 1 \le i \le k,\ 1 \le j \le d.$$

The first terms in the updates, $\bar{y}_t^i \bar{x}_t^j$ and $\bar{y}_t^i x_t^j$, are quite similar. The main difference between the updates is between the second terms: $a_t^i x_t^j$ and $\bar{y}_t^i w_{ij}$. In our network, the second term $a_t^i x_t^j$ serves to whiten the inputs in our network, whereas Földiák's second term $\bar{y}_t^i w_{ij}$ is added as a decay to ensure the weights remain bounded. In addition, our network includes lateral weights $M_{ij}$ which ensure that the projections $y_t^i$ are distinct, and such lateral weights are not included in Földiák's network. While the updates are similar in some respects, it is difficult to compare the outputs of the networks because Földiák's network is postulated rather than derived from a principled objective function, so the network must be simulated numerically in order to evaluate its output.

## 6   Experiments

To validate our approach, we test Bio-SFA (Alg. 1) on synthetic and naturalistic datasets. We provide an overview of the experiments here and defer detailed descriptions and additional figures to Sec. B of the supplement. The evaluation code is available at `https://github.com/flatironinstitute/bio-sfa`.

To measure the performance of our algorithm, we compare the "slowness" of the projection $\mathbf{Y} = \mathbf{M}^{-1}\mathbf{W}\mathbf{X}$, with the slowest possible projection. This can be quantified using the objective (6). We first evaluate the objective (6) at its optimum:

$$\lambda_{\text{slow}} := \min\left\{\operatorname{Tr}\mathbf{V}^\top \mathbf{C}_{\dot{x}\dot{x}}\mathbf{V} : \mathbf{V} \in \mathbb{R}^{m \times k} \text{ s.t. } \mathbf{V}^\top \mathbf{C}_{xx}\mathbf{V} = \mathbf{I}_k\right\}$$

which can be evaluated using an offline generalized eigenvalue problem solver. To compute the error at each iteration, we compare the slowness of the current projection to the minimal slowness:

$$\text{Error} = \tilde{\mathbf{V}}^\top \mathbf{C}_{\dot{x}\dot{x}}\tilde{\mathbf{V}} - \lambda_{\text{slow}}, \qquad \tilde{\mathbf{V}} := \mathbf{W}^\top \mathbf{M}^{-1}(\mathbf{M}^{-1}\mathbf{W}\mathbf{C}_{xx}\mathbf{W}^\top \mathbf{M}^{-1})^{-1/2}, \qquad (19)$$

where the normalization ensures that $\tilde{\mathbf{V}}$ satisfies the constraint in Eq. (6). In Sec. B, we show that $\mathbf{V}$ indeed asymptotically satisfies the constraint in Eq. (6).

### 6.1   Chaotic time series

Before testing on naturalistic datasets, we test Bio-SFA on a challenging synthetic dataset. Let $\{\gamma_t\}$ be a (slow) driving force equal to the sum of 6 sine functions with random amplitudes, frequencies and phases, Fig. 2a (red line). Define the noisy series derived from the recursive logistic map with time-varying growth rate: $z_t = (3.6 + 0.4\gamma_t)z_{t-1}(1 - z_{t-1})$, Fig. 2b (black dots). Wiskott [30] showed that the driving force $\{\gamma_t\}$ can be recovered from the noisy series $\{z_t\}$ by implementing (offline) Quadratic SFA on the 4-dimensional signal $\{\mathbf{s}_t\}$ whose components correspond to the values of the noisy series over the 4 most recent time steps, i.e., $\mathbf{s}_t := (z_t, z_{t-1}, z_{t-2}, z_{t-3})$. We replicate the results from [30] using Bio-SFA. Let $\{\mathbf{x}_t\}$ be the 14-dimensional quadratic expansion of $\{\mathbf{s}_t\}$. We use Bio-SFA to extract the slowest one-dimensional projection $\{y_t\}$, Fig. 2c (green dots). Qualitatively, we see that the slowest projection recovered by Bio-SFA closely aligns with the slow driving force $\{\gamma_t\}$. In Fig. 2d we plot the error at each iteration.

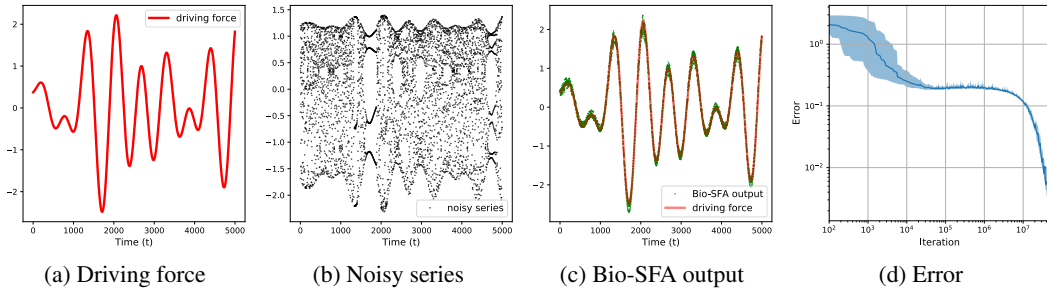

| (a) Driving force | (b) Noisy series | (c) Bio-SFA output | (d) Error |

Figure 2: Performance of Bio-SFA on a noisy series generated by a logistic map with slow driving force. Panels (a), (b) and (c) depict the final 5000 time steps (out of $5 \times 10^7$ time steps) of the normalized driving force $\{\gamma_t\}$ (red line), noisy series $\{z_t\}$ (black dots), and Bio-SFA output $\{y_t\}$ (green dots). Panel (d) shows the mean error and 90% confidence intervals over 10 runs.

## 6.2 Sequence of natural images

Next, we test Bio-SFA on a sequence of natural images. First, a 256-dimensional sequence $\{\mathbf{z}_t\}$ was generated by moving a $16 \times 16$ patch over 13 natural images from [12] via translations, zooms, and rotations, Fig. 3a. To extract relevant features, we follow the exact same procedure as Berkes and Wiskott [1], but replace the offline SFA solver with Bio-SFA to generate a 49-dimensional output signal $\{\mathbf{y}_t\}$. To visualize the 49-dimensional output, we calculate the unit vector $\mathbf{z} \in \mathbb{R}^{256}$ that maximizes $y^i$, for $i = 1, \ldots, 49$. These optimal stimuli, $\mathbf{z}$, which are displayed as $16 \times 16$ patches in Fig. 3b, resemble Gabor patches and are in qualitative agreement with physiological characteristics of complex cells in the visual cortex. This aligns with the results in [1]; see also, [2]. To evaluate the performance of Bio-SFA, we plot the error at each iteration in Fig. 3c.

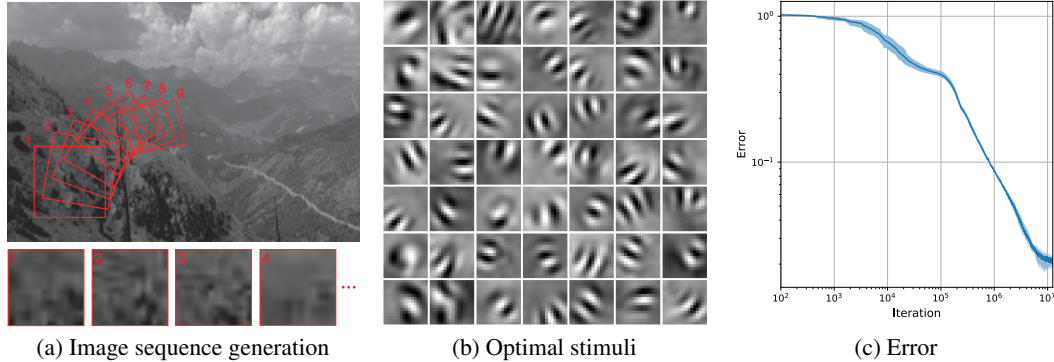

| (a) Image sequence generation | (b) Optimal stimuli | (c) Error |
|---|---|---|

Figure 3: Performance of Bio-SFA on a sequence of natural images. Panel (a) illustrates the generation of the sequence. Panel (b) shows the maximally excitatory stimuli for the 49-dimensional output obtained by Bio-SFA. Panel (c) depicts the mean error and 90% confidence intervals over 10 runs.

## 6.3 Hierarchical SFA on the visual stream of a simulated rat

Following Schönfeld and Wiskott [25], we test a hierarchical 3-layer organization of Bio-SFA "modules" on the inputs from the RatLab framework [24], which simulates the field of view of a rat with random trajectories in a rectangular room. Each layer consists of spatially distributed modules that receive overlapping patches of either the visual stream or the preceding layer. Inside each module, there are 3 steps: (1) Bio-SFA first reduces the dimension of the inputs to generate a 32-dimensional signal, (2) the reduced signal is quadratically expanded, and (3) Bio-SFA reduces the expanded signal to the slowest 32 features. The layers are organized so that the modules in each successive layer receive inputs from larger patches of the visual field, Fig. 4a. Adopting the procedure in [25], the

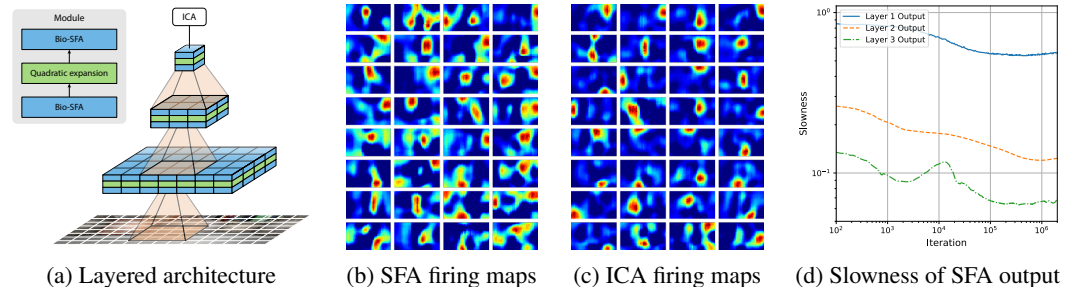

| (a) Layered architecture | (b) SFA firing maps | (c) ICA firing maps | (d) Slowness of SFA output |
|---|---|---|---|

Figure 4: Performance of hierarchical Bio-SFA on a visual stream of a simulated rat. Panel (a) displays a schematic of the layered architecture and the operations within each module. Panels (b) and (c) depict the firing maps of the units in the final SFA layer and the subsequent ICA layer. Each rectangle shows the response of a component of the output as a function of the simulated rat's position within the rectangular room. Panel (d) shows the slowness of each layer's output at each iteration for a single trial.

network is trained greedily layer-by-layer with weight sharing across modules in each layer (see Sec. B of the supplement). The final layer consists of a single module, with a 32-dimensional output, whose spatially-dependent firing maps are shown in Fig. 4b. The 3 SFA layers are followed by a fourth layer, which performs sparse coding via Independent Component Analysis (ICA) [11] (in the offline setting) with a 32-dimensional output, whose firing map is shown in Fig. 4c. As in [25], the firing maps of the final ICA layer are spatially localized and resemble the firing maps of place cells in the hippocampus. To quantify the performance of this hierarchical network, we plot the slowness (not errors, see Sec. B of the supplement) of each of the first 3 layers' outputs at each iteration, Fig. 4d.

## 7  Discussion

We derived an online algorithm for SFA with a biologically plausible neural network implementation, which is an important step towards understanding how the brain could use temporal slowness as a computational principle. While our network implementation satisfies natural requirements for biological plausibility, it differs from biological neural circuits in a number of ways. For instance, our network includes direct lateral inhibitory synapses between excitatory neurons, whereas inhibition is typically modulated by interneurons in biological networks. By adapting the approach in [21], interneurons can be introduced to modulate inhibition. Second, the synaptic updates in our network require both the pre- and post-synaptic neurons to store slow variables; however, signal frequencies in dendrites are slower than in axons, suggesting that it is more likely for slow variables to be stored in the post-synaptic neuron, not the pre-synaptic neuron. We can address this with a modification, which is exact when the expanded signal $\{\mathbf{x}_t\}$ exhibits time-reversibility, so that only the post-synaptic represents slow variables; see Sec. C of the supplement. Finally, our network includes linear neurons, which do not respect the nonnegativity constraints of neuronal outputs. An interesting future direction is to understand the effect of enforcing a nonnegativity constraint on $\mathbf{y}_t$ in the objective function (9).

## Broader impact

An important problem in neuroscience is to understand the computational principles the brain uses to process information. Progress on this front has the potential to have wide ranging benefits for helping to manage the adverse effects of neurological diseases and disorders. This work represents a small step in that direction.

## Acknowledgements

This work was internally support by the Simons Foundation. We thank Yanis Bahroun, Nicholas Chua, Shiva Farashahi, Johannes Friedrich, Alexander Genkin, Jason Moore, Anirvan Sengupta and Tiberiu Tesileanu for helpful comments and feedback on an earlier draft of this work.

## Footnotes

[2]The zeroth time step is included to ensure the discrete-time derivative is defined at $t = 1$.

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
