[Supplementary Material]

## Supplemental material

## A  Detailed derivation of the SFA objective

Here, we provided a detailed derivation of the SFA objective (9). We allow for the case that $\mathbf{C}_{xx}$ is not full rank (but is at least rank $k$).

Our starting point is the objective in Eq. (6), which we recall here:

$$\underset{\mathbf{V}\in\mathbb{R}^{m\times k}}{\arg\min} \ \mathrm{Tr}\,\mathbf{V}^{\top}\mathbf{C}_{\dot{x}\dot{x}}\mathbf{V} \quad \text{subject to} \quad \mathbf{V}^{\top}\mathbf{C}_{xx}\mathbf{V} = \mathbf{I}_k. \tag{20}$$

Under the whitening constraint $\mathbf{V}^{\top}\mathbf{C}_{xx}\mathbf{V} = \mathbf{I}_k$, we have the following relation:

$$\mathbf{V}^{\top}\mathbf{C}_{\dot{x}\dot{x}}\mathbf{V} = 2\mathbf{I}_k - \frac{1}{T}\sum_{t=1}^{T}\mathbf{V}^{\top}(\mathbf{x}_t\mathbf{x}_{t-1}^{\top} + \mathbf{x}_{t-1}\mathbf{x}_t^{\top})\mathbf{V} = 4\mathbf{I}_k - \mathbf{V}^{\top}\mathbf{C}_{\bar{x}\bar{x}}\mathbf{V}.$$

Since $\mathrm{Tr}\,\mathbf{I}_k = k$ is constant, it does not affect the output of the argmin. Therefore, we can rewrite the objective in Eq. (20) as the following maximization problem:

$$\underset{\mathbf{V}\in\mathbb{R}^{m\times k}}{\arg\max} \ \mathrm{Tr}\,\mathbf{V}^{\top}\mathbf{C}_{\bar{x}\bar{x}}\mathbf{V} \quad \text{subject to} \quad \mathbf{V}^{\top}\mathbf{C}_{xx}\mathbf{V} = \mathbf{I}_k. \tag{21}$$

Next, let $k \le n \le m$ be the rank of $\mathbf{C}_{xx}$. We first project $\mathbf{X}$ onto its $n$-dimensional principal subspace; i.e., onto the subspace spanned by eigenvectors of $\mathbf{C}_{xx}$ corresponding to positive eigenvalues. To this end, consider the eigendecomposition $\mathbf{C}_{xx} = \mathbf{U}\mathbf{\Lambda}\mathbf{U}^{\top}$, where $\mathbf{\Lambda}$ is an $n \times n$ diagonal matrix whose diagonal elements are the positive eigenvalues of $\mathbf{C}_{xx}$ and $\mathbf{U}$ is a $m \times n$ matrix whose orthonormal column vectors are the corresponding eigenvectors. Then $\mathbf{U}\mathbf{U}^{\top} \in \mathbb{R}^{m\times m}$ is the matrix that projects $\mathbf{X}$ onto its $n$-dimensional principal subspace and Eq. (21) is equivalent to the maximization problem:

$$\underset{\mathbf{V}\in\mathbb{R}^{m\times k}}{\arg\max} \ \mathrm{Tr}\,\mathbf{V}^{\top}\mathbf{U}\mathbf{U}^{\top}\mathbf{C}_{\bar{x}\bar{x}}\mathbf{U}\mathbf{U}^{\top}\mathbf{V} \quad \text{subject to} \quad \mathbf{V}^{\top}\mathbf{U}\mathbf{U}^{\top}\mathbf{C}_{xx}\mathbf{U}\mathbf{U}^{\top}\mathbf{V} = \mathbf{I}_k. \tag{22}$$

Setting $\hat{\mathbf{x}}_t := \mathbf{\Lambda}^{-1/2}\mathbf{U}^{\top}\bar{\mathbf{x}}_t$ for $t = 1,\ldots,T$, and

$$\hat{\mathbf{V}} := \mathbf{\Lambda}^{1/2}\mathbf{U}^{\top}\mathbf{V}, \qquad \mathbf{C}_{\hat{x}\hat{x}} := \frac{1}{T}\sum_{t=1}^{T}\hat{\mathbf{x}}_t\hat{\mathbf{x}}_t^{\top} = \mathbf{\Lambda}^{-1/2}\mathbf{U}^{\top}\mathbf{C}_{\bar{x}\bar{x}}\mathbf{U}\mathbf{\Lambda}^{-1/2},$$

we see that $\hat{\mathbf{V}}$ is the solution of:

$$\underset{\hat{\mathbf{V}}\in\mathbb{R}^{m\times k}}{\arg\max} \ \mathrm{Tr}\,\hat{\mathbf{V}}^{\top}\mathbf{C}_{\hat{x}\hat{x}}\hat{\mathbf{V}} \quad \text{subject to} \quad \hat{\mathbf{V}}^{\top}\hat{\mathbf{V}} = \mathbf{I}_k. \tag{23}$$

Eq. (23) is the variance maximization objective for the PCA eigenproblem, which is optimized when the column vectors of $\hat{\mathbf{V}}$ span the $k$-dimensional principal subspace of $\mathbf{C}_{\hat{x}\hat{x}}$.

Finally, define the data matrices

$$\bar{\mathbf{X}} := [\bar{\mathbf{x}}_t,\ldots,\bar{\mathbf{x}}_T], \qquad \hat{\mathbf{X}} := [\hat{\mathbf{x}}_1,\ldots,\hat{\mathbf{x}}_T], \qquad \bar{\mathbf{Y}} := [\bar{\mathbf{y}}_1,\ldots,\bar{\mathbf{y}}_T].$$

Then, since $\bar{\mathbf{y}}_t = \mathbf{V}^{\top}\mathbf{U}\mathbf{U}^{\top}\bar{\mathbf{x}}_t = \hat{\mathbf{V}}^{\top}\hat{\mathbf{x}}_t$, we see that $\bar{\mathbf{Y}}$ is the projection of $\hat{\mathbf{X}}_t$ onto its $k$-dimensional principal subspace. As shown in [6], this principal projection can be expressed as a solution of the following objective from classical multidimensional scaling:

$$\underset{\bar{\mathbf{Y}}\in\mathbb{R}^{k\times T}}{\arg\min} \ \frac{1}{2T^2}\big\|\bar{\mathbf{Y}}^{\top}\bar{\mathbf{Y}} - \hat{\mathbf{X}}^{\top}\hat{\mathbf{X}}\big\|_{\mathrm{Frob}}^2 = \underset{\bar{\mathbf{Y}}\in\mathbb{R}^{k\times T}}{\arg\min} \ \frac{1}{2T^2}\big\|\bar{\mathbf{Y}}^{\top}\bar{\mathbf{Y}} - \bar{\mathbf{X}}^{\top}\mathbf{C}_{xx}^{+}\bar{\mathbf{X}}\big\|_{\mathrm{Frob}}^2,$$

where we have used the fact that $\hat{\mathbf{X}}^{\top}\hat{\mathbf{X}} = \bar{\mathbf{X}}^{\top}\mathbf{U}\mathbf{\Lambda}^{-1}\mathbf{U}^{\top}\bar{\mathbf{X}} = \bar{\mathbf{X}}^{\top}\mathbf{C}_{xx}^{+}\bar{\mathbf{X}}$. Lastly, we note that the minimization problem in Eq. (10) is equivalent to the min-max problem in Eq. (11) with $\mathbf{C}_{xx}^{+}$ in place of $\mathbf{C}_{xx}^{-1}$. This can be verified by differentiating $L(\mathbf{W},\mathbf{M},\bar{\mathbf{Y}})$ with respect to $\mathbf{W}$ and noting that the optimal value is achieved when $\mathbf{W}$ equals $\frac{1}{T}\bar{\mathbf{Y}}\bar{\mathbf{X}}^{\top}\mathbf{C}_{xx}^{+}$.

# B  Experimental methods

In this section, we detail how we implemented each experiment.

## B.1  Implementation of neural dynamics

In Bio-SFA, to compute the output $\mathbf{y}_t = \mathbf{M}^{-1}\mathbf{a}_t$, we use the recursive updates $\mathbf{y}_t \leftarrow \mathbf{y}_t + \gamma(\mathbf{a}_t - \mathbf{M}\mathbf{y}_t)$ because they respect the neural architecture. For purposes of simulation, we multiply $\mathbf{M}^{-1}$ by $\mathbf{a}_t$ to compute the output. When $k > 1$, to speed up simulations, we store the value of $\mathbf{M}^{-1}$ and make rank-1 updates at each iteration using the Sherman-Morrison formula.

## B.2  Chaotic time series

**Driving force:**  The driving force $\{\gamma_t\}$ is defined to be the sum of 6 sine functions, as follows:

$$\gamma_t := \sum_{i=1}^{6} A_i \sin(\theta_i t + \omega_i), \qquad t = 1, 2, \ldots.$$

Here the amplitudes $A_1, \ldots, A_6$ are uniformly sampled from the interval $(.1, 2)$ and then normalized so that they sum to 1, the frequencies $\theta_1, \ldots, \theta_6$ are uniformly sampled from the interval $(0.25, 1.25)$, and the phases $\omega_1, \ldots, \omega_6$ are uniformly sampled from the interval $(0, 2\pi)$.

**Hyperparameters:**  We used the learning rate $\eta_t = 1/(a + bt)$. To choose our hyperparameters, we performed a grid search over $a \in \{10^2, 10^3, 10^4, 10^5\}$, $b \in \{10^{-1}, 10^{-2}, 10^{-3}, 10^{-4}\}$ and $\tau \in \{0.01, 0.05, 0.1, 0.5, 1, 5\}$. We found the optimal hyperparameters to be $a = 4$, $b = -4$ and $\tau = 0.5$.

**Hardware:**  The experiment was performed on a 2.8 GHz Quad-Core Intel Core i7 CPU.

## B.3  Sequence of natural images

**Implementation:**  We extracted 2,500 image sequences from the 13 images used in [12]. To generate each sequence, one of the 13 images was chosen uniformly at random, and then a sequence of 100 $16 \times 16$ patches was extracted following the procedure in [1, 2], using the default parameters from the code released with those papers. In particular, sums of sinusoids with random phases and amplitudes are used to drive the translation, zoom and rotation of the $16 \times 16$ field of view. Following [1, 2], we first project the 254-dimensional image sequence onto its 64-dimensional principal subspace. Bio-SFA is then trained on the 2144-dimensional quadratic expansion of the 64-dimensional projected sequence.

**Hyperparameters:**  We used the learning rate $\eta_t = \alpha/(1 + t/\beta)$. To choose our hyperparameters, we performed a grid search over $\alpha \in \{2.5 \times 10^{-6}, 5 \times 10^{-6}, 2.5 \times 10^{-5}, 5 \times 10^{-5}, 2.5 \times 10^{-4}, 5 \times 10^{-4}\}$, $\beta \in \{10^4, 10^6, 5 \times 10^6, 10^7, 5 \times 10^7, 10^8, 10^9, 10^{10}\}$ and $\tau \in \{0.5, 1, 2, 4\}$. We found the optimal hyperparameters to be $\alpha = 5 \times 10^{-6}$, $\beta = 5 \times 10^6$ and $\tau = 1$.

**Orthonormality constraint:**  To evaluate if Bio-SFA satisfies the orthonormality constraint $\mathbf{V}^\top \mathbf{C}_{xx} \mathbf{V} = \mathbf{I}_k$ in Eq. (7), where $\mathbf{V} = \mathbf{W}^\top \mathbf{M}^{-1}$, we use the normalized squared Frobenius norm, defined as follows:

$$\text{Constraint error} = \frac{1}{k} \left\| \mathbf{M}^{-1}\mathbf{W}\mathbf{C}_{xx}\mathbf{W}^\top\mathbf{M}^{-1} - \mathbf{I}_k \right\|_{\text{Frob}}^2. \qquad (24)$$

In Fig. 5, we plot the constraint error at each iteration.

**Hardware:**  The experiment was performed on an NVIDIA Tesla V100 GPU.

## B.4  Hierarchical SFA on the visual stream of a simulated rat

**Simulated visual stream:**  To generate the input data to the hierarchical network, a sequence of 10,000 samples from the default scene in RatLab [24] was generated, following the open field

Figure 5: Convergence of the constraint error defined in Eq. (24) for Bio-SFA (Alg. 1). The lines and shaded regions show the mean error and 90% confidence intervals over ten runs.

experiments in [25]. RatLab simulates a rat's motion by driving its linear and angular momentum by random signals chosen to match experimental data. A wide image is extracted to match the rat's wide field of view. The resulting image sequence is used directly as training data for the online experiments here, after centering and rescaling.

**Architecture:** The hierarchical organization consists of 3 layers of Bio-SFA "modules", described below, followed by a fourth ICA layer; see Fig. 4a. The input to the layered architecture is a sequence of $320 \times 40$ color images. The output of all SFA layers and the ICA layer are sequences of 32-dimensional vectors.

**Description of the layers:** The 4 layers are as follows:

1. The first layer consists of a 2-dimensional array of $63 \times 9$ Bio-SFA modules. Each module receives as input $10 \times 8$ pixel patches sampled from the $320 \times 40$ input, with each patch offset from its neighbors by half of the receptive field width in each dimension. The patches are then transformed into $240 = 80 \times 3$-dimensional vectors to be passed into the modules. The output of each module is a sequence of 32-dimensional vectors.

2. The second layer consists of a 2-dimensional array of $8 \times 2$ Bio-SFA modules. Each module receives inputs from a $14 \times 6$ grid of modules from the first layer, again overlapping each other by half their length in each dimension. Since the output of each module in the first layer is 32-dimensional, the vectorized input to each module has dimension $2688 = 32 \times 14 \times 6$. The output of each module in the second layer is a sequence of 32-dimensional vectors.

3. The third layer consists of a single Bio-SFA module that receives input from all $8 \times 2$ modules in the second layer. Thus, the input to the third layer module has dimension $512 = 32 \times 8 \times 2$. The output of the third layer is a sequence of 32-dimensional vectors.

4. The fourth layer is an offline ICA algorithm, described below. It receives as input the 32-dimensional vector output of the third layer and produces a 32-dimensional output.

**Description of a Bio-SFA module:** Each module receives a sequence of vector inputs (whose dimension depends on the layer) and outputs a 32-dimensional sequence. The module consists of 3 steps:

1. Bio-SFA is applied to the input sequence to generate the slowest 32-dimensional projection.

2. The projected sequence is quadratically expanded to generate the 560-dimensional expanded sequence, which is centered in the online setting using the running mean.

3. Bio-SFA is applied to the expanded sequence to generate a 32-dimensional output.

**Description of the ICA layer:** After an online hierarchy is trained, it is exported to a hierarchy of MDP (Modular toolkit for Data Processing) [35] nodes that can be read by the RatLab framework [24]. Then, RatLab is instructed to fit an ICA layer in the offline setting, using MDP's implementation of CuBICA [4].

**Training procedure:** Following [9], the layers were trained in a greedy layer-wise fashion, i.e., the layers are trained sequentially and the weights in a layer are fixed once it has been trained. The Bio-SFA layers are trained using weight sharing; that is, each layer uses the same synaptic weights $\mathbf{W}$ and $\mathbf{M}$, which are shared across all patches. To compute the $\mathbf{W}$ and $\mathbf{M}$ updates at each training step, the updates for each patch is computed according to Alg. 1, and these updates are summed to generate the updates for $\mathbf{W}$ and $\mathbf{M}$, which are scaled by the square root of the number of patches.

We use time-dependent learning rates of the form $\eta_t = \alpha/(1 + t/\beta)$, with $\beta$ fixed to $5 \times 10^6$ in all modules. For the first Bio-SFA step in the first module, we set $\alpha = 5 \times 10^{-7}$ and $\tau = 5 \times 10^{-4}$. In the rest of the modules, the first Bio-SFA step used $\alpha = 2.5 \times 10^{-6}$ with the same $\tau$. For the second Bio-SFA step in each module, we set $\alpha = 5 \times 10^{-5}$ and $\tau = 1$.

**Firing maps:** To generate a firing map from either the final SFA layer or the ICA layer, RatLab is instructed to generate a test set of still images by sampling the visual field of the simulated rat across a fine grid of spatial positions, using 8 head directions at each location. The output activities from each of the 32 units in either the final SFA or ICA layer are averaged over head orientation to generate a heatmap of that unit's activity over the spatial grid. Those maps are shown in Fig. 4.

**Quantification of slowness:** To demonstrate that each layer is finding slower features, we plot the "slowness" of each layer's output, which is defined by

$$\text{Slowness} = \tilde{\mathbf{V}}^\top \mathbf{C}_{\dot{x}\dot{x}} \tilde{\mathbf{V}},$$

where $\tilde{\mathbf{V}}$ is defined as in Eq. (19) and $\mathbf{C}_{\dot{x}\dot{x}}$ denotes the covariance of the discrete-time derivative of the expanded input for that module.

**Hardware:** This experiment was performed on an Intel Xeon Gold 6148 CPU.

## C SFA for reversible processes

The update for $\mathbf{W}$ in Alg. 1 requires the input neurons to store both the input, $\mathbf{x}_t$, and the delayed sum, $\bar{\mathbf{x}}_t$. Here, we propose a modification of the algorithm, which is exactly SFA in the case that the expanded input $\{\mathbf{x}_t\}$ is reversible, that only requires the input neurons to store the input $\mathbf{x}_t$. Suppose the expanded signal $\{\mathbf{x}_t\}$ exhibits time-reversal symmetry; that is,

$$\frac{1}{T}\sum_{t=1}^{T}\mathbf{x}_t\mathbf{x}_{t-1}^\top = \frac{1}{T}\sum_{t=1}^{T}\mathbf{x}_{t-1}\mathbf{x}_t^\top.$$

Then

$$\begin{aligned}
\mathbf{C}_{\bar{x}\bar{x}} &= \frac{1}{T}\sum_{t=1}^{T}\bar{\mathbf{x}}_t(\mathbf{x}_t + \mathbf{x}_{t-1})^\top \\
&= \frac{1}{T}\sum_{t=1}^{T}\bar{\mathbf{x}}_t\mathbf{x}_t^\top + \frac{1}{T}\sum_{t=1}^{T}(\mathbf{x}_t + \mathbf{x}_{t-1})\mathbf{x}_{t-1}^\top \\
&= \frac{1}{T}\sum_{t=1}^{T}\bar{\mathbf{x}}_t\mathbf{x}_t^\top + \frac{1}{T}\sum_{t=1}^{T}\mathbf{x}_{t-1}\mathbf{x}_t^\top + \frac{1}{T}\sum_{t=1}^{T}\mathbf{x}_t\mathbf{x}_t^\top + \frac{1}{T}(\mathbf{x}_0\mathbf{x}_0^\top - \mathbf{x}_T\mathbf{x}_T^\top) \\
&= \frac{2}{T}\sum_{t=1}^{T}\bar{\mathbf{x}}_t\mathbf{x}_t^\top + \frac{1}{T}(\mathbf{x}_0\mathbf{x}_0^\top - \mathbf{x}_T\mathbf{x}_T^\top) \\
&= 2\mathbf{C}_{\bar{x}x} + \frac{1}{T}(\mathbf{x}_0\mathbf{x}_0^\top - \mathbf{x}_T\mathbf{x}_T^\top),
\end{aligned}$$

where

$$\mathbf{C}_{\bar{x}x} := \frac{1}{T}\sum_{t=1}^{T}\bar{\mathbf{x}}_t\mathbf{x}_t^\top.$$

In the large $T$ limit, we can approximate the offline gradient descent update for $\mathbf{W}$ in Bio-SFA by replacing $\mathbf{C}_{\bar{x}\bar{x}}$ with $2\mathbf{C}_{\bar{x}x}$, which results in the update

$$\mathbf{W} \leftarrow \mathbf{W} + 2\eta(2\mathbf{M}^{-1}\mathbf{W}\mathbf{C}_{\bar{x}x} - \mathbf{W}\mathbf{C}_{xx}).$$

Recalling that $\bar{\mathbf{y}}_t = \mathbf{M}^{-1}\mathbf{W}\bar{\mathbf{x}}_t$, we can write the online stochastic gradient descent step for $\mathbf{W}$ as

$$\mathbf{W} \leftarrow \mathbf{W} + 2\eta(2\bar{\mathbf{y}}_t - \mathbf{a}_t)\mathbf{x}_t^\top.$$

This yields our online SFA algorithm for reversible processes (Alg. 2).

---

**Algorithm 2:** Bio-SFA for reversible processes

---

**input** expanded signal $\{\mathbf{x}_0, \mathbf{x}_1, \ldots, \mathbf{x}_T\}$; dimension $k$; parameters $\gamma, \eta, \tau$
**initialize** matrix $\mathbf{W}$ and positive definite matrix $\mathbf{M}$
**for** $t = 1, 2, \ldots, T$ **do**
  $\mathbf{a}_t \leftarrow \mathbf{W}\mathbf{x}_t$
  **repeat**
    $\mathbf{y}_t \leftarrow \mathbf{y}_t + \gamma(\mathbf{a}_t - \mathbf{M}\mathbf{y}_t)$               ▷ compute output
  **until** convergence
  $\bar{\mathbf{x}}_t \leftarrow \mathbf{x}_t + \mathbf{x}_{t-1}$
  $\bar{\mathbf{y}}_t \leftarrow \mathbf{y}_t + \mathbf{y}_{t-1}$
  $\mathbf{W} \leftarrow \mathbf{W} + 2\eta(\bar{\mathbf{y}}_t - \mathbf{a}_t)\mathbf{x}_t^\top$         ▷ stochastic gradient descent-ascent steps
  $\mathbf{M} \leftarrow \mathbf{M} + \frac{\eta}{\tau}(\bar{\mathbf{y}}_t\bar{\mathbf{y}}_t^\top - \mathbf{M})$
**end for**

---

As with Bio-SFA, Alg. 2 can be implemented in the neural network shown in Fig. 6. Note that in this case, the elementwise synaptic update for $W_{ij}$, given by

$$W_{ij} \leftarrow W_{ij} + 2\eta(2\bar{y}_t^i - a_t^i)x_t^j,$$

depends only on $\bar{y}_t^i$, $a_t^i$ and $x_t^j$, so the pre-synaptic input neuron only needs to represent the $x_t^j$, as opposed to both $x_t^j$ and $\bar{x}_t^j$. Biologically, this is more realistic because the signal frequency of dendrites is slower than the signal frequency of axons, so it is more likely that slow variables are represented in the post-synaptic neuron.

| Variable | Biological interpretation |
|---|---|
| $\mathbf{x}_t$ | expanded signal |
| $\mathbf{W}$ | feedforward synaptic weights |
| $\mathbf{a}_t := \mathbf{W}\mathbf{x}_t$ | dendritic current |
| $\mathbf{M}$ | lateral synaptic weights |
| $\mathbf{y}_t$ | output signal |

| Neural dynamics & plasticity rules |
|---|
| $d\mathbf{y}_t(\gamma)/d\gamma = \mathbf{a}_t - \mathbf{M}\mathbf{y}_t(\gamma)$ |
| $\Delta\mathbf{W} = 2\eta(2\mathbf{y}_t + 2\mathbf{y}_{t-1} - \mathbf{a}_t)\mathbf{x}_t^\top$ |
| $\Delta\mathbf{M} = \frac{\eta}{\tau}((\mathbf{y}_t + \mathbf{y}_{t-1})(\mathbf{y}_t + \mathbf{y}_{t-1})^\top - \mathbf{M})$ |

Figure 6: A biologically plausible neural network implementation of Bio-SFA for reversible processes. The figure on the left depicts the architecture of the neural network. Blue circles are the input neurons and black circles are the output neurons with separate dendritic and somatic compartments. Lines with circles connecting the neurons denote synapses. Filled (resp. empty) circles denote excitatory (resp. inhibitory) synapses.

We test Alg. 2 on the naturalistic image sequences from [28], which are not reversible (due to the rotation of the images). In Fig. 7a, we display the optimal stimuli for the filters that are found by Alg. 2. These optimal stimuli are in close qualitative agreement with the optimal stimuli found by Bio-SFA, shown in Fig. 3. In Fig. 7b, we plot the error defined in Eq. (19) and find that Alg. 2 (Bio-SFA for reversible processes) performs comparably with Alg. 1 (Bio-SFA).

(a) Optimal Stimuli

(b) Error

Figure 7: Performance of Bio-SFA for reversible processes on a sequence of natural images. Panel (a) shows the maximally excitatory stimuli for the 49-dimensional output obtained by Bio-SFA for reversible processes. Panel (b) shows the mean error and 90% confidence intervals over 10 runs.