[Reviews · NeurIPS 2020]

Review 1

Summary and Contributions: Derives a biologically plausible version of SFA, and runs it on a standard moving image datasets, obtaining results similar to previous batch learning methods.

Strengths: An elegant formulation of the problem and nice solution.

Weaknesses: Perhaps could benefit from stronger ties to potential neural mechanisms, or where in the brain you may expect to find this, experimental predictions, etc. It seems this is the main point after all. Also, how do you propose that biology computes the nonlinear expansion \mathbf{x}_t?

Correctness: yes as far as I can tell

Clarity: Yes.

Relation to Prior Work: The development in section 2.1 and 2.2 seems straight of Wiskott's papers, but you wouldn't really know that from reading the text. Its cited by numbers, but I think highlighting more directly in the text would be helpful.

Reproducibility: Yes

Additional Feedback: I find the paper interesting, but somehow not super interesting. It's never seemed an issue to me that you could have a biologically plausible version of SFA. Foldiak's original model was formulated that way after all. So not sure what this development adds in that context. I'm sure there advantages to this approach, but it would be good to bring them out more.


Review 2

Summary and Contributions: The paper develops a slow feature analysis (SFA) method that is biologically plausible, in that it operates online and has weight updates that only use information that is available in the presynaptic and postsynaptic neurons. The method is tested on several datasets, and it is shown that cost is reduced with training and that interesting receptive fields emerge.

Strengths: - I agree with the authors that a biologically plausible implementation of SFA is an important development. - The development of the method from standard SFA was systematic and rigorous.

Weaknesses: - Some limitations were mentioned in the discussion. Of these, the fact that the method uses linear neurons seems to be the most serious. I appreciate that the authors defined their sense of "biologically plausible", but linear neurons don't sit very well with this term.

Correctness: Yes, it seems to be correct.

Clarity: The writing is clear and well organized. The math was clearly and efficiently communicated.

Relation to Prior Work: Yes.

Reproducibility: Yes

Additional Feedback: - I had regrettably not checked many of the equations in the first round, but I have checked most of them now. - Thank you for addressing my minor concern about the assumption of full rank in Line 108. I look forward to the updated explanation in the final version of the paper.


Review 3

Summary and Contributions: This paper produces a so-called biologically plausible neural network for slow-feature analysis. Biological plausibility here means that network learning is online and based on local synaptic learning rules. These online and locality requirements might lead to low computational overhead. While Foldiak, Wiskott, and many others have explored online local learning for SFA in the last thirty years, this paper attempts to relate SFA to a normative theory through an MDS objective.

Strengths: While Foldiak, Wiskott, and others have explored local online learning for SFA in the last thirty years, this paper successfully relates it to the normative MDS objective. Similar work on biologically plausible implementations has been one-dimensional; this work extends that to multiple dimensions.

Weaknesses: The conceptual and theoretical innovations are limited, which is not surprising given that the problem has been worked on for the last thirty years, most notably by Wiskott's lab. Claim of biological plausibility seems weak, limited only to local learning rules and online learning.

Correctness: The basic idea and the logic seem to be sound, though I have not scrutinized all the equations.

Clarity: Reasonably clear and well written.

Relation to Prior Work: This work seems to be a marriage of Wiskott's line of research and Pehlevan and Chklovskii's recent approach.

Reproducibility: Yes

Additional Feedback: I read all the reviews and rebuttal, and can see the potential contribution relative to existing work better. I keep my score. It is an acceptable paper.


Review 4

Summary and Contributions: While SFA is a classical and widely used unsupervised learning algorithm, a neural plausible neural algorithm for SFA is still missing. This paper presents a solution and the numerical results well support the proposed method. So overall, it's a very useful piece of work.

Strengths: 1. The presented method is very useful to the computational neuroscience community. 2. The paper presents a reformulation of SFA, which leads to a neural plausible SFA algorithm. 3. The numerical results seem to be solid and well support the proposed theory.

Weaknesses: There are some minor typos and notation issues, just like most of the NeurIPS drafts. If accepted for publication, the authors should definitely make an effort to polish the paper.

Correctness: I checked most of the derivation and I think they are correct. Except there are a few minor typos, e.g. the equation above equation (7), V and V^T are missing. The optimization above equation (8), a transpose is missing. Things like these should be fixed but do not bother me much.

Clarity: Section 2 is too long, some of the unnecessary review shall be moved to the appendix. Section 3 shall be expanded and discuss the major results more thoroughly.

Relation to Prior Work: This piece of works has been missing for a long time. I think it can fill the gap.

Reproducibility: Yes

Additional Feedback:

[Author Response · NeurIPS 2020]

We thank the reviewers for their careful reading of our work and for their helpful comments.

**Relation to Foldiak and Wiskott et al.'s work and advantages & relevance of a normative approach [R1, R3]:**
Our paper builds on work by Foldiak [8], who proposed local learning rules to capture relevant temporal correlations, but resorted to numerical simulation to compute the output of the network; and work by Wiskott et al., who proposed SFA as a spectral analysis approach for capturing temporal correlations (or equivalently, slowness) and can be readily computed in the offline setting, but for which there was no efficient multi-channel biologically plausible network implementation. Our normative approach allows us to derive an algorithm with local learning rules directly from the spectral data analysis problem, providing a new perspective on the works of both Foldiak and Wiskott et al. and clarifying their relationship. If accepted, we will expand this discussion in the main text and point out the similarities (and differences) between our learning rules and Foldiak's [8]. We will also clarify that the text in sections 2.1 and 2.2 is essentially a review of work by Wiskott et al.

**Relation to biology and experimental evidence [R1]:** While we do not map our neural network onto a specific circuit in the brain, the general neural architecture of feedforward and lateral connections is ubiquitous. The main experimental support is from Wiskott et al. who have shown that SFA reproduces response properties of complex cells in the visual cortex and hierarchical SFA reproduces properties of place cells in the hippocampus.

In terms of experimental predictions, our work predicts the synaptic weights in the SFA circuit. For example, it predicts that the singular values of $\mathbf{W}\mathbf{C}_{\bar{x}\bar{x}}\mathbf{W}^\top$ are equal to the singular values of $\mathbf{M}^2$, where $\mathbf{W}$ are the weights of the feedforward connections, $\mathbf{M}$ are the weights of the lateral connections and $\mathbf{C}_{\bar{x}\bar{x}}$ is the covariance matrix of the summed time-lagged signal. With the increasing progress in connectomics, it may be possible to estimate $\mathbf{W}$ and $\mathbf{M}$, which could provide evidence for or against a circuit implementing our proposed SFA network.

With regards to the nonlinear expansion, experimental evidence suggests that the cortex performs (sparse) signal expansions; see, e.g., (DeWeese, Wehr and Zador, 2003) and (Olshausen and Field, 2004). There are various proposed mechanisms for performing sparse codings, including (Olshausen and Field, 1997) and (Arora, Ge, Ma and Moitra, 2014). One mechanism for implementing a *quadratic* expansion are so-called "Sigma-Pi units" (Rumelhart, Hinton and McClelland, 1986), which use gating to implement multiplication of inputs and have been invoked in cortical modeling (Mel and Koch, 1990).

**Full rank assumption of the expanded signal [R2]:** We assume that the expanded signal $\{\mathbf{x}_t\}$ has full rank for mathematical convenience (namely, so we can take an inverse of the covariance matrix). In reality, the expanded signal may not be full rank. One approach is to first perform PCA on the expanded signal to reduce its dimension; however, this would add a layer to the network. Alternatively, one can simply replace the inverse of the covariance matrix $\mathbf{C}_{xx}^{-1}$ in the objective (see Eq. 8) with its Moore-Penrose pseudo inverse $\mathbf{C}_{xx}^+$ (provided the rank of the expanded signal is at least $k$). In this case, the derivation proceeds exactly as laid out in the paper. We will include a detailed justification in the appendix that this substitution in fact achieves the same objective.

**Typos and organization [R4]:** Thank you for pointing out the typos. We will correct them and carefully read the paper for other typos and grammatical mistakes. In addition, we will move some previously known technical details to the appendix and expand the discussion of our derivation.

[Meta-Review · NeurIPS 2020]

All reviewers agreed that the proposed normative approach to deriviing a biologically plausible implementation of SFA in multiple dimensions represents a strong contribution.